# New Reports of *Phytophthora* Species in Plant Nurseries in Spain

**DOI:** 10.3390/pathogens11080826

**Published:** 2022-07-23

**Authors:** Beatriz Mora-Sala, Maela León, Ana Pérez-Sierra, Paloma Abad-Campos

**Affiliations:** 1Instituto Agroforestal Mediterráneo, Universitat Politècnica de València, Camino de Vera s/n, 46022 Valencia, Spain; mleonsan@eaf.upv.es (M.L.); pabadcam@eaf.upv.es (P.A.-C.); 2Forest Research, Alice Holt Lodge, Farnham GU10 4LH, UK; ana.perez-sierra@forestresearch.gov.uk

**Keywords:** woody plants, ornamental plants, plants for planting, oomycetes, *Phytophthora*, invasive pathogens

## Abstract

The plant nursery industry has become an ideal reservoir for *Phytophthora* species and other soilborne pathogens. In this context, isolation from tissues and soil of ornamental and forest plants from nurseries in four regions of Spain was carried out. A high diversity of *Phytophthora* species was confirmed. Fourteen *Phytophthora* phylotypes (*P. cactorum*, *P. cambivora*, *P. cinnamomi*, *P. citrophthora*, *P. crassamura*, *P. gonapodyides*, *P. hedraiandra*, *P. nicotianae*, *P. niederhauserii*, *P. palmivora*, *P. plurivora*, *P. pseudocryptogea*, *P. sansomeana,* and *Phytophthora* sp. tropicalis-like 2) were isolated from over 500 plant samples of 22 species in 19 plant genera. Nine species were detected in water sources, two of them (*P. bilorbang* and *P. lacustris*) exclusively from water samples. *P. crassamura* was detected for the first time in Spain. This is the first time *P. pseudocryptogea* is isolated from *Chamaecyparis lawsoniana* and *Yucca rostrata* in Spain.

## 1. Introduction

Different pests and diseases can affect nursery production, which can turn plants into pathogen vectors [1]. Pathogens affect all plant industry sectors across agriculture, horticulture, forestry, and amenity, and they can have a significant impact on yield, market access, sustainability of production, food security, and product integrity [2]. Fungi is considered the kingdom with the largest number of phytopathogenic species [3]. In addition to this, the kingdom Straminipila embraces other important plant pathogens such as *Phytophthora* and *Pythium* [3]. *Phytophthora* species are responsible for large losses of nursery stock throughout the world [4,5,6,7,8,9,10,11].

*Phytophthora* is one of the most destructive genera which includes currently over 150 known species and about 100 more that are in the process of being described [11,12,13,14]. Almost all *Phytophthora* species are ecologically and economically important plant pathogens worldwide, some of them with a broad host range. *Phytophthora* species possess wide environmental adaptation that ranges from terrestrial to aquatic habitats. Some species, such as *P**hytophthora infestans* (Mont.) de Bary, have been responsible for some of the most important epidemics in history, others such as *Phytophthora cinnamomi* Rands and *Phytophthora ramorum* Werres, de Cock and Man in ’t Veld, disrupt and diminish biodiversity in natural ecosystems [15,16,17,18,19,20,21]. Others such as *Phytophthora citrophthora* (R.E. Smith and E.H. Smith) Leonian, *Phytophthora nicotianae* Breda de Haan, *Phytophthora hedraiandra* de Cock and Man in ’t Veld, *Phytophthora niederhauserii* Z.G. Abad and J.A. Abad and *P. ramorum*, produce major losses in the nursery industry worldwide [7,11,14,22,23,24,25,26,27].

The inocula of *Phytophthora* spp., which cause foliar as well as root diseases, can increase from low to high levels within a few days or weeks under favourable conditions [28]. Polycyclic diseases can turn into serious epidemics when environmental conditions favour a rapid production of *Phytophthora* propagules [5]. The movement of plants and plant products between biogeographical zones due to human activity constitutes the leading pathway for the introduction of pathogens and exotic pests [3,29,30].

*Phytophthora* in its centre of origin does not necessarily constitute an ecological problem or even noticeable because the binomial pathogen–host has co-evolved [28]. *Phytophthora* native hosts have developed specific defences which confer some tolerance against the pathogen [11,29,31,32]. Nevertheless, when the pathogen is transferred to a new habitat with favourable conditions, it can likely extend to a wide range of new hosts causing serious ecological and economical losses [33,34]. The arrival of new genotypes, lineages, or exempt mating types into a non-native habitat can pose an additional risk to the ecosystem and possibly drive host range expansion for that species [11,23,29,33,34].

Invasive pathogens have been causing damage to native plant communities, woodlands, and landscapes on a global scale for over a century [29,35]. Nursery trade encourages, unintentionally, the dispersal and establishment of invasive and exotic *Phytophthora* spp. [3,11,36,37]. Even more, the high specialisation and intensification of nursery production favours the reproduction and hybridisation of invasive species enhancing the dispersion and settlement of these on natural ecosystems [29]. The diversity of the genus has increased rapidly in the last decades due to the appearance of new alien species such as *Phytophthora alni* subsp. *alni* Brasier and Kirk, *Phytophthora austrocedri* Greslebin and Hansen, *Phytophthora foliorum* Donahoo and Lamour, *P. hedraiandra*, *Phytophthora kernoviae* Brasier, Beales and Kirk, *Phytophthora lateralis* Tucker and Milbrath, *P. pinifolia* Durán, Gryzenh. and Wingf., *Phytophthora pluvialis* Reeser, Sutton and Hansen or *P. ramorum*, which requires routine samplings for their early detection, and due to the numerous surveys on unexplored habitats, such as water reservoirs [34,38].

In Spain, since the first report of *P. ramorum* in 2002 [39,40], surveys have been carried out in ornamental nurseries, garden centres, public gardens, and forest masses to detect and eradicate this pathogen. These surveys have shown that other species of *Phytophthora* affect many ornamental plants, posing a risk also to nurseries and natural ecosystems [9,41].

Due to the increasing threat of invasive *Phytophthora* species, and the high risk of hybridisation, a survey was carried out on producing woody and/or ornamental plant nurseries to investigate the presence of *Phytophthora* and soilborne fungi present and to exclude the presence of quarantine pathogens. Therefore, the aim of this investigation was to identify *Phytophthora* species and fungal pathogens in ornamental/forest nurseries in different geographical areas of Spain, which could be a threat to the plant nursery industry and to managed and natural ecosystems.

## 2. Materials and Methods

### 2.1. Study Sites

Surveys were conducted in 25 Spanish nurseries located in four geographically different regions during the period 2012–2014 in Catalonia (provinces of Barcelona, Girona, Tarragona, and Lleida), Comunidad Valenciana (provinces of Alicante, Castellón, and Valencia), Extremadura (province of Cáceres), and Basque Country (province of Guipúzcoa) (Figure 1).

In ornamental nurseries, only symptomatic plants were collected, whereas in forest nurseries for habitat restoration non-symptomatic plants were also collected. Foliar symptoms (leaf blotch, blight, chlorosis, defoliation), wilting, dieback, growth reduction, cankers with or without gummosis, rot and presence of dead plants were considered symptoms associated to possible *Phytophthora* or soilborne pathogen infection (Figure 2). These plants presented in most cases root rot and/or loss of the feeder roots with the presence of necrotic lesions (Figure 2). Plant samples were collected together with their pot media or soil, individually stored in labelled plastic bags, and kept in cold conditions until they were processed in the laboratory at the Instituto Agroforestal Mediterráneo, Universitat Politècnica de València (IAM-UPV). In total, 78 samples were collected from 10 nurseries in Catalonia, 343 samples from 5 nurseries in Comunidad Valenciana, 110 samples from 8 nurseries in Extremadura, and 16 samples from 2 nurseries in Basque Country. Each sample consisted of one plant.

Thirteen water samples from recirculating irrigation ponds were also collected during the survey in Catalonia and from one nursery located in the Comunidad Valenciana. Ten litres of water were filtered using three cellulose membranes (5.0 µm pore diameter, Millipore Corporation), which were placed in sterile Petri dishes, sealed with parafilm, labelled, and stored in a cooler during transport to the laboratory. Furthermore, while water sampling was being performed, two additional samples consisting of five leaves showing *Phytophthora*-like spots floating in two of the surveyed water ponds, were collected in Catalonia; these were labelled and transported for processing in the laboratory.

### 2.2. Isolation from Plant Tissues, Soil, and Water

Plant samples (leaves and/or roots) were separated from substrate media, and the roots were washed and kept for 24 h in tap water that was repeatedly renewed for oxygenation. Samples were superficially disinfected spraying alcohol at 70% for oomycete isolation and disinfected for 1 min in a 1.5% sodium hypochlorite solution and washed twice with sterile distilled water for fungi isolation [25,42]. Small fragments from the lesion edge were plated on semi-selective media for isolation of oomycetes (CMA-PARPB supplemented or not with hymexazol [43]). Plates were incubated at 20 °C in the dark for 3–5 days for fungi and up to 7 days for oomycetes. All the colonies grown on isolation media were transferred to PDA plates and incubated at 20 °C in darkness for 7 days for identification. Pure cultures of all putative *Phytophthora* isolates were obtained by transferring single hyphal tips to PDA plates.

The soil removed from each plant sample was baited using Granny Smith apples targeting oomycetes species isolation [5]. Four 10 mm-diameter and 1–1.5 cm-deep holes were made on the apple fruit with a cork borer, each one was filled with the soil sample, saturated with distilled water, sealed with adhesive tape, and incubated at room temperature until lesions appeared (4–7 days). Small tissue fragments from lesion edges were plated on CMA-PARPB with and without hymexazol and incubated at 20 °C in darkness. Each colony was transferred to PDA and incubated as described above for plant samples.

Oomycetes isolation from filtered water samples was undertaken also by apple baiting; three longitudinal flap-like cuts (one per membrane filter) were made on a Granny Smith apple. In each flap cut, half of the subsample membrane was placed, sealed with parafilm, and incubated at room temperature until symptoms develop (4–7 days). The re-isolation from the apple was performed from the edge of any lesions that developed after incubation following the protocol described above.

### 2.3. Identification

#### Molecular Identification

DNA from *Phytophthora* and *Pythium* isolates was extracted from pure cultures grown on PDA by scraping the mycelium and mechanically disrupting it by grinding to a fine powder under liquid nitrogen, using the EZNA Plant Miniprep Kit (Omega Bio-tek, Doraville, GE, USA) following the manufacturer’s instructions.

Nuclear ribosomal DNA ITS amplifications were carried out using the universal primers ITS4 and ITS6 that target conserved regions in the 18S and 28S rDNA genes [44,45]. All PCR reactions were performed using HotBegan™ Taq DNA Polymerase (Canvax Biotech SL, Córdoba, Spain), according to the manufacturer’s instructions, in a PTC 200 thermo-cycler (MJ Research, Waltham, MA, USA) with the following parameters: 94 °C for 3 min; 35 cycles of 94 °C for 30 s, 55 °C for 30 s, and 72 °C for 45 s; and 72 °C for 10 min. Amplified products were purified and sequenced by Macrogen (Amsterdam, The Netherlands).

The isolates were identified to the species level by conducting Basic Local Alignment Search Tool (BLAST) searches with the sequence data on international collection databases (*Phytophthora* Database, PhyID, and GenBank) and a customised database that included the new *Phytophthora* species described and segregated from *Phytophthora* complexes and provisional taxons. An isolate was assigned to a species when the identity was above the 99% cut-off in respect to the ex-type isolates. The ITS sequence did not resolve the identity of 10 isolates. Therefore, for these isolates the mitochondrial cytochrome c oxidase I (COI) region was amplified using the primers OomCoxI-Levup and Fm85mod [46].

The DNA sequences from this study (Table 1), together with those of reference species of each clade retrieved from Genbank, were aligned using the ClustalW algorithm [47] contained within the MEGA X software package [48]. The sequences of the reference isolates were selected from ex-type or well-authenticated *Phytophthora* species recommended in IDphy: molecular and morphological identification of *Phytophthora* (https://idtools.org/id/phytophthora/molecular.php (accessed on 15 February 2021).). The alignments were inspected and corrected manually. Incomplete portions at either end of the alignments were excluded prior to analyses.

Phylogenetic analyses were based on Bayesian inference (BI), maximum likelihood (ML), and maximum parsimony (MP). Bayesian analyses were performed using MrBayes v 3.2.6 on the NGPhylogeny.fr web service [49]. Four simultaneous analyses were run for 100,000 generations, sampling every 10,000, with four Markov chain Monte Carlo (MCMC) chains. The first 25% of saved trees were discarded and posterior probabilities were determined from the remaining trees. The ML analyses were completed with the tool Randomized Axelerated Maximum Likelihood (RAxML) implemented on the T-REX web server (http://www.trex.uqam.ca/ (accessed on 11 July 2022).) [50]. ML tree searches were performed under the generalised time-reversible with gamma correction (GTR + Γ) nucleotide substitution model using 1000 pseudoreplicates. The other parameters were used as default settings. MP analyses were performed in MEGA X [48] with the Tree Bisection and Reconnection (TBR) algorithm, where gaps were treated as missing data. The robustness of the topology was evaluated using 1000 bootstrap replications [51]. Measures for the maximum parsimony such as tree length (TL), consistency index (CI), retention index (RI), and rescaled consistency index (RC) were also calculated.

### 2.4. Conservation of Phytophthora and Pythium Isolates

Pure cultures obtained by hyphal tipping were maintained in the oomycete culture collection at the IAM-UPV. Each isolate was grown on V-8 Juice Agar and incubated at 20 °C for 7 days in darkness. A total of 15 mycelium plugs (6 mm diameter) from the border of the colony were extracted and placed into a 12 cm^3^ glass flask which contained 1.5% sterile soil extract solution for long-term conservation at 14 °C. The sterile soil extract solution was prepared mixing 100 g of soil with 900 mL distilled water. The mixture was stirred and allowed to stand for 24 h. Subsequently, 50 mL of the supernatant was taken and added to 950 mL of distilled water to be autoclaved.

*Phytophthora* isolates were also conserved in tubes with Oat Agar medium (72.5 g L^−1^ oatmeal agar, Sigma Aldrich, Steinheim, Germany) for long-term storage. A single 6 mm-diameter agar disk was placed in each OA tube, incubated at 25 °C until mycelium growth was observed, and then it was sealed with parafilm for conservation at 14 °C.

## 3. Results

### 3.1. Symptomatology

In all nurseries, a broad range of symptoms was observed: cankers (with or without gummosis exudates), collar rot, dead plants, dieback (partial dieback or the whole plant), foliar symptoms (chlorosis, defoliation, leaf spots, irregular shaped blotches in the leaf margins or starting at the leaf apex or petiole, necrotic spots), growth reduction, and wilting (Figure 2). Figure 3 shows the percentage distribution of symptoms observed in the sampled plants collected in the nurseries referred to the total number of plants collected (in blue colour) and to the number of plants on which *Phytophthora* was isolated (in green colour). The most frequent symptoms among the total number of collected samples were dieback (37.1%), followed by foliar symptoms (29.4%) and growth reduction (13.3%).

A total of 547 samples were collected and oomycetes were identified in 30.7% of the plant samples. The most frequent symptoms observed in samples positive for oomycetes were dieback (43.5%), foliar symptoms (28.6%), and growth reduction (11.3%).

*Phytophthora* was isolated from 59 plants (Table 2), which means 10.8% of total plant samples collected in this survey. On plants affected by *Phytophthora*, dieback was the most frequent symptom observed (59.3%), followed by foliar symptoms (27.1%), wilting, and growth reduction (both 6.8%) (Figure 3). In almost all positive *Phytophthora* plants, the aerial symptomatology corresponded with a damaged root system. Nevertheless, in some plants, the damage was limited to the aerial part, with no visible root symptoms. Furthermore, in only two non-symptomatic plants *Phytophthora* was isolated.

### 3.2. Phytophthora Species Isolated in the Study

Seventy-one isolates of *Phytophthora* were recovered from 18 nurseries from the four locations (Table 2). These isolates were obtained from infected tissues (roots) and/or the rhizosphere soil of 547 plant samples belonging to 22 species included in 19 plant genera (Table 2). Thirty-six *Phytophthora* isolates were isolated from water samples collected in Catalonia region from the irrigation ponds (Table 2).

Molecular identification of the isolates revealed the presence of 17 *Phytophthora* phylotypes (Figure 4). The ITS alignment consisted of 887 positions including gaps. Of these, 568 were constant and 270 were parsimony-informative characters. The heuristic search using MP generated the 10 most parsimonious trees (TL = 603, CI = 0.651, RI = 0.934, RC = 0.608), from which one was selected (Figure 4). The topology of the phylogenetic tree inferred by MP analysis was identical to those obtained by the BI and ML analyses; therefore, only the MP tree is presented with MP and ML bootstrap support values and BI posterior probability scores at the nodes (available on request). Sequences from this study were deposited in Genbank (Table 1).

The species isolated were *Phytophthora bilorbang* Aghighi and Burgess, *Phytophthora cactorum* (Lebert and Cohn) J. Schr., *Phytophthora cambivora* (Petri) Buisman, *P. cinnamomi*, *P. citrophthora*, *Phytophthora crassamura* Scanu, Deidda and Jung, *P. hedraiandra*, *Phytophthora gonapodyides* (H.E. Petersen) Buisman, *Phytophthora lacustris* Brasier, Cacciola, Nechwatal, Jung and Bakonyi, *P. nicotianae*, *P. niederhauserii*, *Phytophthora palmivora* E.J. Butler, *Phytophthora plurivora* T. Jung and T.I. Burgess, *Phytophthora pseudocryptogea* Safaiefarahani, Mostowfizadeh, Hardy and Burgess and *Phytophthora sansomeana* Hansen and Reeser. Two *Phytophthora* isolates recovered from the roots of *Arbutus unedo* and *Juniperus communis* were identified as the informally designated taxon *Phytophthora* sp. tropicalis-like 2 [52]. There were four *Phytophthora* isolates that could not be identified to the species level, so they were tentatively named as *Phytophthora* sp. 1 clade 2.

The ITS of *Phytophthora* sp. 1 clade 2 was closely related to *Phytophthora meadii* McRae showing differences in two positions with the ex-type, whereas the COI results placed these isolates close to *P. citrophthora*. Therefore, these isolates were designated as *Phytophthora* sp. 1 clade 2.

Among plant and soil samples, *P. pseudocryptogea* was the species with the highest incidence (21.1%), followed by *P. plurivora* (15.5%), *P. hedraiandra* (9.9%), *P. citrophthora* and *P. nicotianae* (8.5% each species), *P. cactorum*, *P. cinnamomi*, and *Phytophthora* sp. 1 clade 2 (5.6% each species), *P. crassamura* and *P. sansomeana* (4.2% each species), *P. gonapodyides*, *P. palmivora*, and *Phytophthora* sp. tropicalis-like 2 (2.8% each species). Two other species, *P. cambivora* and *P. niederhauserii*, had the lowest incidence values (1.4% each species). Some plants were co-infected with more than one *Phytophthora* species. Mixed infections occurred on *Chamaecyparis lawsoniana* (*P. plurivora*–*P. psedocryptogea*), *Citrus sinensis* (*P. citrophthora*–*Phytophthora* sp. 1 clade 2), *Cupressus sempervirens* (*P. palmivora*–*P. plurivora*), *Escallonia* sp. (*P. citrophthora*–*P. nicotianae*), *Juniperus communis* (*P. gonapodyides*–*Phytophthora* sp. tropicalis-like 2), *Pistacia lentiscus* (*P. nicotianae*–*P. palmivora*), *Quercus ilex* (*P. plurivora*–*P. pseudocryptogea* and *P. pseudocryptogea*–*P. sansomeana*), and *Rosmarinus officinalis* (*P. citrophthora*–*P. nicotianae*).

In the aquatic habitats, *P. cactorum*, *P. lacustris*, and *P. gonapodyides* were the most abundant species (26.3%, 21.1%, and 18.4% respectively), followed by *P. cambivora*, *P. citrophthora*, and *P. plurivora* (7.9% each species), *P. pseudocryptogea* (5.3%). The lowest value of presence in water was 2.6%, shared by *P. bilorbang* and *P. palmivora*.

### 3.3. Pythium and Phytopythium Species Isolated in the Study

In total, 6 *Phytopythium* species (*Pp. chamaehyphon*, *Pp. helicoides*, *Pp. litorale*, *Pp. mercuriale*, *Pp. montanum*, and *Pp. vexans*) and 11 *Pythium* species (*Py. sterilum*, *Py. intermedium*, *Py. attrantheridium*, *Py. rostratifingens*, *Py. oopapillum*, *Py. irregulare*, *Py. ultimum*, *Py. undulatum*, *Py. sylvaticum*, *Py. Pleroticum*, and *Py. diclinum*) were isolated in the survey.

## 4. Discussion

This study provides evidence of *Phytophthora*’s wide spread in ornamental and forest nurseries, since the pathogen was isolated from plant material and water samples in the large majority of surveyed nurseries.

In the surveyed nurseries, the sampled plants showed crown symptoms that could be associated with *Phytophthora* infection, such as dieback, shoot blight, chlorosis, defoliation, irregular leaf blotches, wilting, and cankers with gummosis. The symptomatology of aerial plant parts was generally associated with root damage such as change in colour, lesions, absence, and/or rot of the feeder roots. This set of observed symptoms agree with the symptomatology described in the literature [9,11,25,27,53,54,55,56]. It should be noted that disease symptoms may be suppressed due to prophylactic fungicide treatments or the natural lag period between root and crown rots and the development of foliar symptoms [26].

Seventy-one *Phytophthora* isolates from clades 1, 2, 4, 6, 7, and 8 were recovered from 22 species included in 19 plant genera. From some of the plants more than one species of *Phytophthora* was isolated, revealing mixed infections as in previous nursery surveys [9,10,33,54,57]. Some species were very frequent (*P. pseudocryptogea*, *P. plurivora*, *P. hedraiandra*, *P. citrophthora*, and *P. nicotianae*) and others were rare (*P. cambivora* and *P. niederhauserii*, as well as the informally designated taxon, *Phytophthora* sp. tropicalis-like 2).

Four *Phytophthora* species represented a significant finding for the Spanish nursery sector. This study is the first report of *P. crassamura* in Spain; *Phytophthora crassamura* sp. nov. was described by Scanu et al. in Sardinia (Italy) [58] and since then it has been isolated from other hosts in Italy and in California [59,60]. As our isolates were baited from the *P. pinea* nursery substrate, we cannot state *P. pinea* as a new *P. crassamura* host, even the three seedlings showed a highly diminished root system with no secondary feeder roots. This finding suggests that probably *P. pinea* seedlings are susceptible to *P. crassamura*. As initially these isolates were misidentified as *Phytophthora megasperma* Dreschsler, no pathogenicity tests were performed. *Phytophthora pseudocryptogea* [61] was reported on *Quercus ilex* in 2018 in different regions of Spain and the present study not only confirms its presence in the nurseries from those regions [62,63] but also it was isolated for the first time in Spain on *Chamaecyparis lawsoniana* and *Yucca rostrata*. Moreover, this is the first time *P. sansomeana* was isolated in Europe and in *Q. ilex* worldwide. *Phytophthora sansomeana* was segregated from the *P. megasperma* complex in 2009 and until now it was only in the United States and in China, from diverse forest and agricultural hosts, such as Douglas-fir nursery seedlings, weeds, and soybean [64,65,66]. Since it is not the first time that the species has been identified in nursery material, its pathogenicity on holm oak should be tested to understand the risk it poses to this fundamental tree species of forest ecosystems and landscapes of Mediterranean Europe. Lastly, two isolates from our study clustered with *Phytophthora* sp. tropicalis-like 2 described by Jung et al. in 2020 based on ITS blast-assigned identity with the isolate VN830 [52]. This is a provisional first report of *Phytophthora* sp. tropicalis-like 2 on *Arbutus unedo* and *Juniperus communis*.

In previous nursery surveys in Spain, *P. cactorum*, *P. cinnamomi*, *P. citricola*, *P. citrophthora*, *P. cryptogea*, *Phytophthora drechsleri* Tucker, *Phytophthora hibernalis* Carne, *Phytophthora multivora* Scott and Jung, *P. nicotianae*, *P. niederhauserii*, *P. palmivora*, *P. plurivora*, *Phytophthora syringae* (Kleb.) Kleb., *Phytophthora tentaculata* Kröber and Marwitz, and *P. tropicalis* have been reported [9,25,67]. According to Moralejo *et al.*, *P. cinnamomi* and *P. cryptogea* (probably *P. pseudocryptogea*) have escaped from nurseries and are currently spreading in *Q. ilex* forests, and infect associated shrubs such as *Arbutus unedo* and *Cistus monspeliensis* in the lowlands of northern Mallorca [25]. Other studies in nurseries worldwide recovered almost the same species which demonstrates that global nursery trade is the main pathway for *Phytophthora* dispersion [6,9,11,25,26,27,29,33,53,54,56,57,60,68,69,70,71,72].

In Europe, a very extensive analysis of incidence of *Phytophthora* spp. was conducted, based on data from 23 countries between 1972 and 2013, in order to study the pathway of *Phytophthora* from nurseries into natural, semi-natural, and horticultural ecosystems [11]. From nursery plant material, 49 *Phytophthora* taxa were identified, being *P. plurivora*, *P. cinnamomi*, *P. cactorum*, *P. nicotianae*, *P. ramorum* and *P. citrophthora* the most commonly sampled species, considered all alien pathogens in Europe. From forest and landscape plantings, 56 *Phytophthora* taxa were recovered, and invasive species with wide host ranges, such as *P. plurivora*, *P. cinnamomi*, *P. nicotianae*, *P. cryptogea*, and *P. cactorum*, were the most common. This large-scale study demonstrates that *Phytophthora* infect nursery stock across Europe and the spread of these pathogens through infested nursery stock into natural ecosystems.

In California, Sims et al. [60] reported *P. cactorum* as the most frequent species in restoration nurseries but *P. hedraiandra*, *P. multivora*, *P. occultans*, *P. crassamura*, *P. thermophila*, and *P. pseudocryptogea* were also isolated. Rooney-Latham et al. [72] reported *P. tentaculata*, *P. cactorum*, *P. cryptogea* complex, *P. cambivora*, *P. cinnamomi*, *P. citricola*, *P. hedraiandra*, *P. megasperma*, *P. multivora*, *P. nicotianae*, *P. niederhauserii*, *P. parvispora*, *P. pini*, *P. plurivora*, and *P. riparia* in Californian nurseries.

Regarding water surveys, the nine species reported in this study once again agree with *Phytophthora* spp. recovered from irrigation water, waterways, or riparian ecosystems published in other studies [73,74,75,76,77,78,79,80,81,82]. It is not surprising that as *Phytophthora* is adapted for aquatic dispersal, multiple *Phytophthora* spp. have been recovered from waterways or irrigation waters. Indeed, several novel species have been detected in the last decade from water fluxes or riparian ecosystems such as *Phytophthora lateralis* (clade 8) causing *Chamaecyparis lawsoniana* decline [83], *Phytophthora alni* (clade 7) causing *Alnus* spp. decline [84], and *P. ramorum* (clade 8) causing sudden oak death on *Quercus* spp. and *Notholithocarpus densiflorus* [17]. Detection of *Phytophthora* taxa belonging to clade 6 has increased in recent years as riparian systems have grown in attention [13,77,85,86]. *Phytophthora* spp. from clade 6 are thought to be adapted to survive in rivers due to their rapid colonisation of leaves and plant debris [87,88]. Jung et al. consider the possibility that species from clade 6 are probable saprotrophs, as these *Phytophthora* spp. depend on their ability to rapidly colonise fresh plant material (such as fallen leaves) in order to outcompete other saprotrophic organisms [88]. There is a significant gap in understanding waterborne plant pathogens, particularly in open irrigation systems [78,82,89].

Among other plant pathogens that were also isolated, the most important genera were *Pythium* and *Phytopythium*. The percentage of recovered *Pythium* and *Phytopythium* species highlights the importance of sanitary measures in the nursery industry. *Pythium* and *Phytopythium* are also among the most frequent plant pathogens in nurseries (seed rot and damping-off), *Pythium* species require free water to complete their cycle but compared with *Phytophthora*, they have a quicker development and growth. Most *Pythium* and *Phytopythium* species used to be considered saprotrophs but nowadays the pathogenicity of some species has been demonstrated [90,91,92,93].

The impact that plant pathogens can have on the plant industry can extend into billions of dollars, but the worst is the environmental risk, which biodiversity, forestry, and agriculture are currently experiencing [11,33,94,95]. Biosecurity needs to be the cornerstone of the global nursery trade to avoid the possibility of *Phytophthora* spp. spreading to new habitats where they may be exposed to compatible species and potentially form new hybrids [29,96,97].

The exclusion of nursery pathogens from forested areas is a critical issue for forest health [60,98]. Monitoring the pathogen zone, restricting vehicle movement from infested to uninfested areas, cleaning vehicles before entering uninfested areas, preventing infested and uninfested soil mixing, preventing water draining from infested to uninfested areas, and education of public and forestry workers are some of the exclusion measures that should come into full force and effect [60,98].

A high priority should be placed on the production of pathogen-free propagating material by appropriate sanitary practices [99]. The microbial community plays an important role in the protective effect against Oomycetes. Organic soils in the form of compost have long been found to supress a number of *Phytophthora* and *Pythium* spp. [99]. Nursery sanitation measures such as the following ought be implemented in all nurseries and garden centres: use of new seedling containers, container media pasteurised; irrigation water *Phytophthora*-free (sand filters or chlorine interventions); water splash kept off leaves and wetness time minimised; containers kept off the ground; suppressive composts or fungicides avoided; sustained heat treatment to kill resting structures in plant or soil material via composting, solarisation, oven treatment or autoclaving, heating installation in greenhouses, correct aeration between seedling benches and plantations, pH control (a low pH [3.5–4.5] to avoid spore liberation), moderate nitrogen fertilisation, and routine tool disinfection [98].

It has long been known that nursery stock is the most common pathway for the introduction of new *Phytophthora* species into natural habitats worldwide [11]. Supplying healthy plants should be the fundamental principle of nursery production. Implementing molecular detection through the most recent, effective, and specific assays for *Phytophthora* [33,62,100] will facilitate early detection and the application of control measures to minimise the risk of spread through plant trade.

## 5. Conclusions

This study confirms the widespread presence of pathogens in plant nursery stocks and the risk posed by the plants for the planting pathway. Seventeen *Phytophthora* phylotypes were isolated from tissues and rizosphere soil of 22 plant species in 19 genera and from water samples. The presence of *Phytophthora* mixed infections is noteworthy. It is also relevant reporting, for the first time, the presence of *P. crassamura*, *P. pseudocryptogea*, *P. sansomeana*, and *Phytophthora* sp. tropicalis-like 2 in the Spanish nursery industry. The need for preventing *Phytophthora* dispersion to natural ecosystems must be translated in implementing new policies at the global scale. Good biosecurity practices in nurseries and early detection are critical to mitigate the risk of spread of these pathogens.

## Figures and Tables

**Figure 1 pathogens-11-00826-f001:**
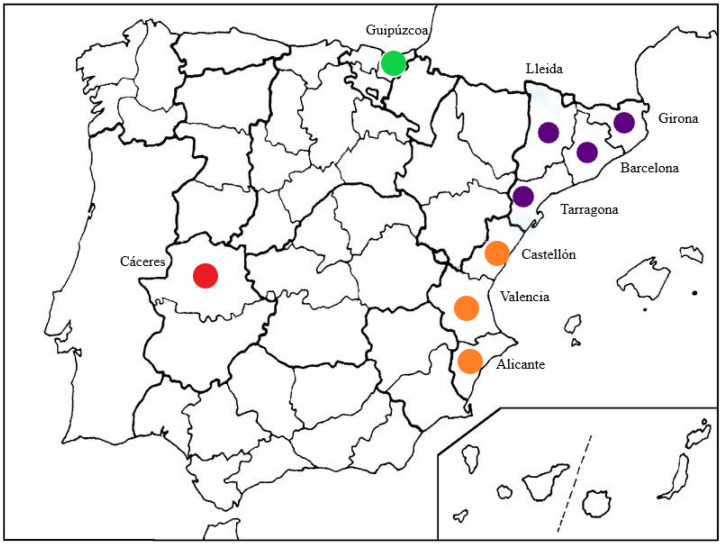
Spanish provinces in which nursery surveys were conducted: in green Basque Country, in purple Catalonia, in orange Comunidad Valenciana, and in red Extremadura.

**Figure 2 pathogens-11-00826-f002:**
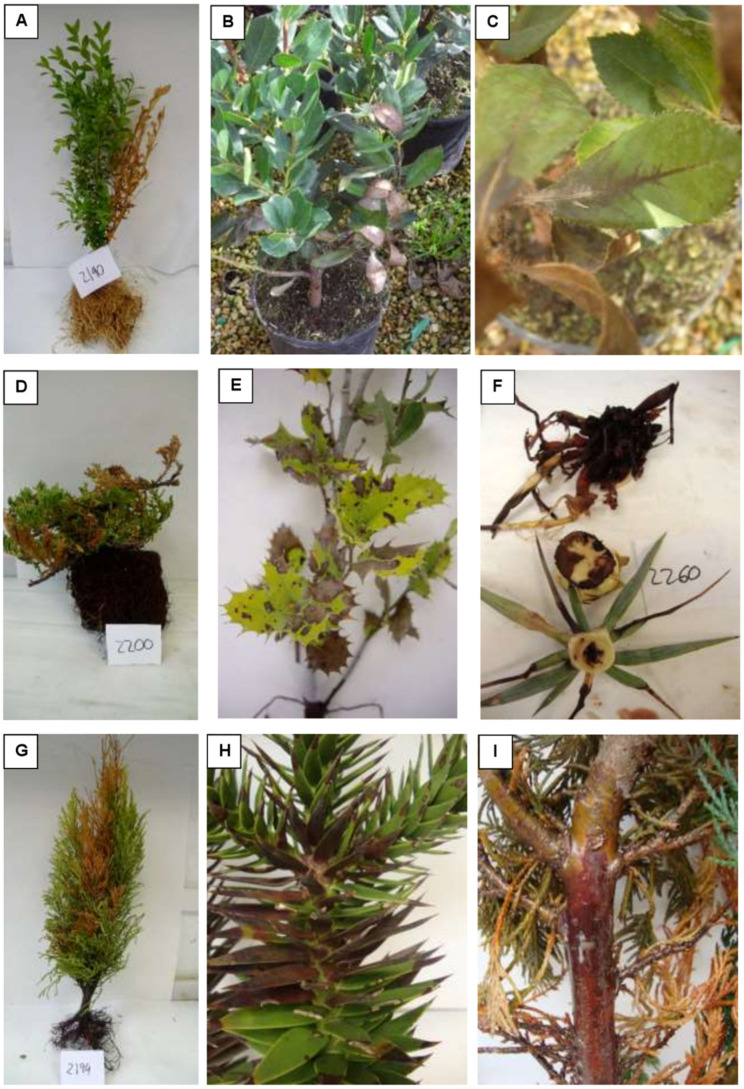
(**A**,**B**,**D**,**G**) Dieback type symptoms on *Buxus sempervirens*, *Arbutus unedo*, *Juniperus horizontalis,* and *Cupressus macrocarpa*. (**C**) *Arbutus unedo* with black leaf necrosis that advances along the middle vein from the petiole to the apex. (**E**) *Quercus ilex* showing chlorosis and leaf spots. (**F**): Rot on *Yucca rostrata*. (**H**) *Araucaria araucana* with leaf necrosis. (**I**) Canker on the stem of a *Chamaecyparis lawsoniana*.

**Figure 3 pathogens-11-00826-f003:**
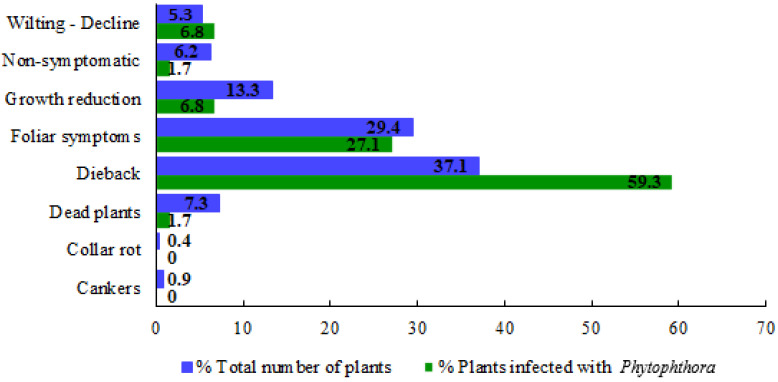
Symptomatology observed in the sampled nurseries expressed in percentage. Bars in blue colour represent the percentage of samples that present a symptomatology regarding the total number of plants of the survey. Bars in green show the symptomatology associated to *Phytophthora* expressed in percentage (number of samples regarding to those plants infected by *Phytophthora*).

**Figure 4 pathogens-11-00826-f004:**
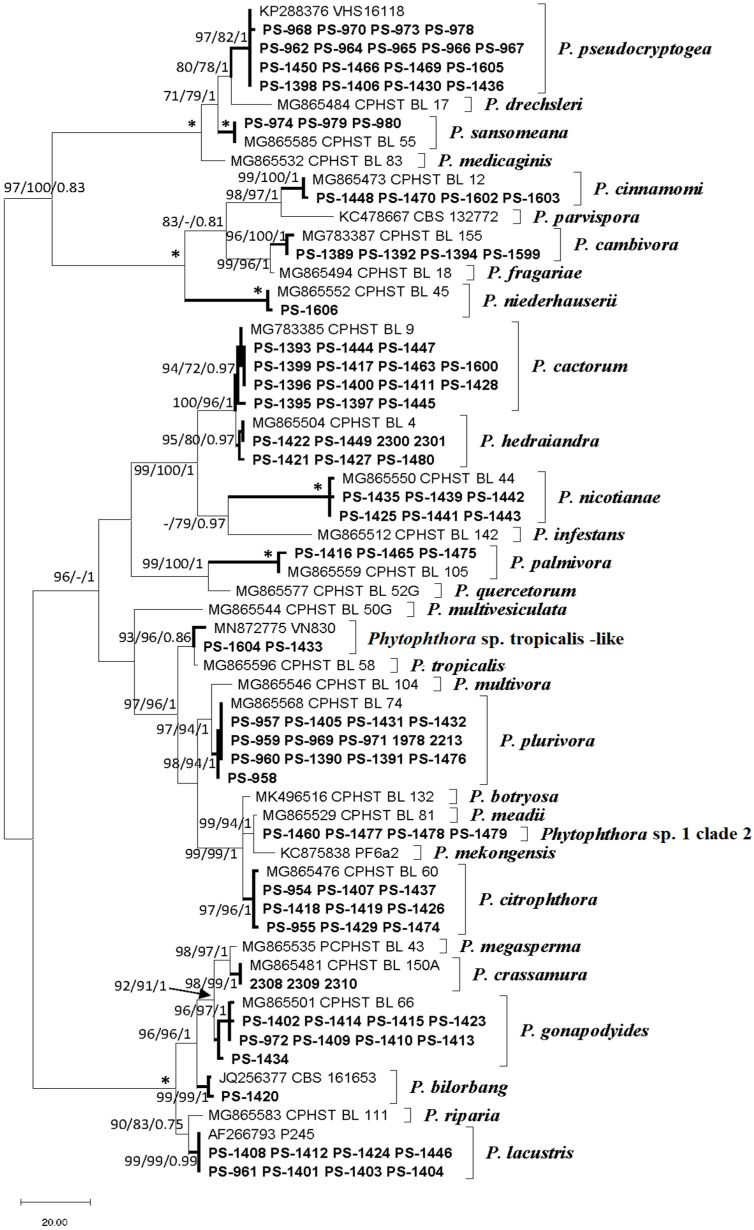
One of 10 most parsimonious trees resulting from the analysis of the internal transcribed spacer (ITS) sequences from isolates. Support values (Maximum Parsimony bootstrap (MP BS)/Maximum Likelihood bootstrap (ML BS)/Bayesian inference posterior probabilities (BI PP)) are given at the nodes. Bootstrap values less than 70% or posterior probabilities less than 0.7 are indicated with “-”. Branches with an asterisk indicate branch support with MP BS = 100%, ML BS = 100%, and BI PP values = 1.0. The scale bar shows the number of substitutions per site. Species identified in the current study are in bold in the clades. The tree was midpoint rooted.

**Table 1 pathogens-11-00826-t001:** *Phytophthora* isolates identified in this study.

Species	Strain Number	Host	Location	GenBank Accession Numbers	
				ITS	COI
*P. bilorbang*	PS-1420	Nursery water pond	Alguaire, Lleida, Spain	MW314333	-
*P. cactorum*	PS-1463	*Photinia x fraseri* “Red Robin” (roots)	St Coloma de Farners, Barcelona, Spain	MW314274	-
	PS-1411	Nursery water pond	St Hilari Sacalm, Girona, Spain	MW314275	-
	PS-1417	Nursery water pond	Alguaire, Lleida, Spain	MW314276	-
	PS-1447	*Photinia x fraseri* “Red Robin” (roots)	St Coloma de Farners, Barcelona, Spain	MW314277	-
	PS-1396	Nursery water pond	St Coloma de Farners, Barcelona, Spain	MW314278	-
	PS-1395	Nursery water pond	St Coloma de Farners, Barcelona, Spain	MW314279	-
	PS-1445	Nursery water pond	St Coloma de Farners, Barcelona, Spain	MW314280	-
	PS-1428	*Photinia x fraseri* “Red Robin” (roots)	St Coloma de Farners, Barcelona, Spain	MW314281	-
	PS-1397	Nursery water pond	St Coloma de Farners, Barcelona, Spain	MW314282	-
	PS-1400	Nursery water pond	St Coloma de Farners, Barcelona, Spain	MW314283	-
*P. cactorum*	PS-1600	*Fagus sylvatica* (soil)	Ataún, Guipúzcoa, Spain	MW314287	-
	PS-1393	Nursery water pond	Fogars de Montclus, Montseny, Barcelona, Spain	MW314286	-
	PS-1399	Nursery water pond	St Coloma de Farners, Barcelona, Spain	MW314285	-
	PS-1444	Nursery water pond	St Coloma de Farners, Barcelona, Spain	MW314284	-
*P. cambivora*	PS-1394	Nursery water pond	St Coloma de Farners, Barcelona, Spain	MW314353	-
	PS-1392	Nursery water pond	Fogars de Montclus, Montseny, Barcelona, Spain	MW314354	-
	PS-1389	Nursery water pond	Fogars de Montclus, Montseny, Barcelona, Spain	MW314356	-
	PS-1599	*Quercus ilex* (roots)	Quart de Poblet, Valencia, Spain	MW314355	-
*P. cinnamomi*	PS-1448	*Arbutus unedo* (roots)	Fogars de Montclus, Montseny, Barcelona, Spain	MW314360	-
	PS-1470	*Arbutus unedo* (roots)	Fogars de Montclus, Montseny, Barcelona, Spain	MW314357	-
*P. cinnamomi*	PS-1602	*Pseudotsuga menziesii* (roots)	Villabona, Guipúzcoa, Spain	MW314358	-
	PS-1603	*Pinus radiata* (roots)	Villabona, Guipúzcoa, Spain	MW314359	-
*P. citrophthora*	PS-1419	Nursery water pond	Alguaire, Lleida, Spain	MW314301	-
	PS-954	*Quercus faginea* (soil)	Pobla de Benifassà, Castellón, Spain	MW314302	-
	PS-1407	Nursery water pond	St Coloma de Farners, Barcelona, Spain	MW314303	-
	PS-1418	Nursery water pond	Alguaire, Lleida, Spain	MW314304	-
	PS-1426	*Escallonia* sp. (soil)	St Coloma de Farners, Barcelona, Spain	MW314305	-
	PS-1429	*Picea pungens* “Glauca Globosa” (roots)	St Hilari Sacalm, Girona, Spain	MW314306	-
	PS-955	*Quercus faginea* (roots)	Pobla de Benifassà, Castellón, Spain	MW314307	-
	PS-1474	*Rosmarinus* sp. (roots)	Mont-Roig del Camp, Tarragona, Spain	MW314308	-
	PS-1437	*Citrus sinensis* (soil)	Alcanar, Tarragona, Spain	MW314309	-
*P. crassamura*	2308	*Pinus pinea* (soil)	Plasencia, Cáceres, Spain	MW314336	-
	2309	*Pinus pinea* (soil)	Plasencia, Cáceres, Spain	MW314335	-
*P. crassamura*	2310	*Pinus pinea* (soil)	Plasencia, Cáceres, Spain	MW314334	-
*P. gonapodyides*	PS-1415	Nursery water pond	St Hilari Sacalm, Girona, Spain	MW314338	-
	PS-1434	*Juniperus hibernicus* (roots)	Alguaire, Lleida, Spain	MW314337	-
	PS-1410	Nursery water pond	St Hilari Sacalm, Girona, Spain	MW314342	-
	PS-1414	Nursery water pond	St Hilari Sacalm, Girona, Spain	MW314339	-
	PS-1423	Leaves floating in nursery water pond	Fogars de Montclus, Montseny, Barcelona, Spain	MW314341	-
	PS-972	*Quercus ilex* (soil)	La Hunde, Ayora, Valencia, Spain	MW314340	-
	PS-1409	Nursery water pond	St Hilari Sacalm, Girona, Spain	MW314344	-
	PS-1402	Nursery water pond	St Coloma de Farners, Barcelona, Spain	MW349609	-
	PS-1413	Nursery water pond	St Hilari Sacalm, Girona, Spain	MW314343	-
*P. hedraiandra*	PS-1449	*Quercus ilex* (soil)	Plasencia, Cáceres, Spain	MW314288	-
	2300	*Quercus ilex* (soil)	Plasencia, Cáceres, Spain	MW314290	-
	2301	*Quercus ilex* (soil)	Plasencia, Cáceres, Spain	MW314293	-
*P. hedraiandra*	PS-1480	*Juniperus phoenicia* (soil)	Camp de Mirra, Alicante, Spain	MW314289	-
	PS-1427	*Viburnum tinus* (collar)	St Coloma de Farners, Barcelona, Spain	MW314291	-
	PS-1422	*Viburnum tinus* (roots)	St Hilari Sacalm, Girona, Spain	MW314292	-
	PS-1421	*Viburnum tinus* (roots)	St Coloma de Farners, Barcelona, Spain	MW314294	-
*P. lacustris*	PS-1408	Nursery water pond	St Coloma de Farners, Barcelona, Spain	MW314345	-
	PS-1424	Leaves floating in nursery water pond	St Coloma de Farners, Barcelona, Spain	MW314346	-
	PS-1412	Nursery water pond	St Hilari Sacalm, Girona, Spain	MW314347	-
	PS-1401	Nursery water pond	St Coloma de Farners, Barcelona, Spain	MW314348	-
	PS-1446	Nursery water pond	St Hilari Sacalm, Girona, Spain	MW314349	-
	PS-961	Nursery fountain	La Hunde, Ayora, Valencia, Spain	MW314350	-
	PS-1404	Nursery water pond	St Coloma de Farners, Barcelona, Spain	MW314351	-
*P. lacustris*	PS-1403	Nursery water pond	St Coloma de Farners, Barcelona, Spain	MW314352	-
*P. nicotianae*	PS-1441	*Pistacia lentiscus* (roots)	St Coloma de Farners, Barcelona, Spain	MW314298	-
	PS-1425	*Escallonia* sp. (roots)	St Coloma de Farners, Barcelona, Spain	MW314299	-
	PS-1439	*Citrus sinensis* (roots)	Alcanar, Tarragona, Spain	MW314297	-
	PS-1435	*Rosmarinus* sp. (roots)	Mont-Roig del Camp, Tarragona, Spain	MW314295	-
	PS-1442	*Myrtus communis* “Tarentina” (roots)	St Coloma de Farners, Barcelona, Spain	MW314296	-
	PS-1443	*Buxus sempervirens* (roots)	St Coloma de Farners, Barcelona, Spain	MW314300	-
*P. niederhauserii*	PS-1606	*Arbutus unedo* (soil)	Plasencia, Cáceres, Spain	MW314361	-
*P. palmivora*	PS-1465	*Pistacia lentiscus* (roots)	St Coloma de Farners, Barcelona, Spain	MW314330	-
	PS-1475	*Cupressus sempervirens* (roots)	Alcanar, Tarragona, Spain	MW314331	-
	PS-1416	Nursery water pond	Alguaire, Lleida, Spain	MW314332	-
*P. plurivora*	1978	*Quercus faginea* (roots)	Pobla de Benifassà, Castellón, Spain	MW314325	-
*P. plurivora*	PS-1405	Nursery water pond	St Coloma de Farners, Barcelona, Spain	MW314314	-
	PS-1432	*Chamaecyparis lawsoniana* “Elwoodii” (soil)	St Hilari Sacalm, Girona, Spain	MW314326	MW314042
	PS-1476	*Cupressus sempervirens* (roots)	Alcanar, Tarragona, Spain	MW314327	MW314043
	PS-957	*Quercus faginea* (soil)	Pobla de Benifassà, Castellón, Spain	MW314315	-
	PS-959	*Quercus faginea* (roots)	Pobla de Benifassà, Castellón, Spain	MW314316	-
	PS-1431	*Chamaecyparis lawsoniana* “Elwoodii” (soil)	St Hilari Sacalm, Girona, Spain	MW314317	-
	PS-1390	Nursery water pond	Fogars de Montclus, Montseny, Barcelona, Spain	MW314318	-
	PS-1391	Nursery water pond	Fogars de Montclus, Montseny, Barcelona, Spain	MW314319	-
	PS-958	*Quercus faginea* (roots)	Pobla de Benifassà, Castellón, Spain	MW314320	-
	PS-960	*Quercus faginea* (roots)	Pobla de Benifassà, Castellón, Spain	MW314321	-
	PS-969	*Juniperus* sp. (soil)	La Hunde, Ayora, Valencia, Spain	MW314322	-
*P. plurivora*	PS-971	*Quercus ilex* (roots)	La Hunde, Ayora, Valencia, Spain	MW314323	-
	2213	*Juniperus chinensis* “Expansa Variegata” (roots)	Fogars de Montclus, Montseny, Barcelona, Spain	MW314324	-
*P. pseudocryptogea*	PS-1469	*Ilex x meserveae* “Blue Maid” (roots)	Fogars de Montclus, Montseny, Barcelona, Spain	MW314362	-
	PS-1450	*Chamaecyparis obtusa* “Nana gracilis” (roots)	Fogars de Montclus, Montseny, Barcelona, Spain	MW314363	-
	PS-1430	*Chamaecyparis lawsoniana* “Elwoodii” (roots)	St Hilari Sacalm, Girona, Spain	MW314364	-
	PS-967	*Quercus ilex* (soil)	La Hunde, Ayora, Valencia, Spain	MW314365	-
	PS-966	*Quercus ilex* (roots)	La Hunde, Ayora, Valencia, Spain	MW314366	-
	PS-1406	Nursery water pond	St Coloma de Farners, Barcelona, Spain	MW314367	-
	PS-1436	*Yucca rostrate* (collar)	Mont-Roig del Camp, Tarragona, Spain	MW314368	-
	PS-1466	*Chamaecyparis lawsoniana* “Elwoodii” (roots)	St Hilari Sacalm, Girona, Spain	MW314369	-
*P. pseudocryptogea*	PS-970	*Quercus ilex* (soil)	La Hunde, Ayora, Valencia, Spain	MW314370	-
	PS-964	*Quercus ilex* (roots)	La Hunde, Ayora, Valencia, Spain	MW314371	-
	PS-978	*Quercus ilex* (soil)	La Hunde, Ayora, Valencia, Spain	MW314372	-
	PS-973	*Quercus ilex* (soil)	La Hunde, Ayora, Valencia, Spain	MW314373	-
	PS-962	*Quercus ilex* (roots)	La Hunde, Ayora, Valencia, Spain	MW314374	-
	PS-965	*Quercus ilex* (roots)	La Hunde, Ayora, Valencia, Spain	MW314376	-
	PS-968	*Quercus ilex* (soil)	La Hunde, Ayora, Valencia, Spain	MW314377	-
	PS-1398	Nursery water pond	St Coloma de Farners, Barcelona, Spain	MW314375	-
	PS-1605	*Quercus ilex* (soil)	Plasencia, Cáceres, Spain	MW314378	-
*P. sansomeana*	PS-974	*Quercus ilex* (soil)	La Hunde, Ayora, Valencia, Spain	MW314379	-
	PS-979	*Quercus ilex* (soil)	La Hunde, Ayora, Valencia, Spain	MW314380	MW314044
*P. sansomeana*	PS-980	*Quercus ilex* (roots)	La Hunde, Ayora, Valencia, Spain	MW314381	MW314045
*Phytophthora* sp. tropicalis-like 2	PS-1433	*Juniperus hibernicus* (soil)	Alguaire, Lleida, Spain	MW314329	MW314040
	PS-1604	*Arbutus unedo* (roots)	Plasencia, Cáceres, Spain	MW314328	MW314041
*Phytophthora* sp. 1 clade 2	PS-1477	*Citrus sinensis* (soil)	Alcanar, Tarragona, Spain	MW314310	MW314036
	PS-1460	*Citrus sinensis* (soil)	Alcanar, Tarragona, Spain	MW314311	MW314037
	PS-1478	*Citrus sinensis* (roots)	Alcanar, Tarragona, Spain	MW314312	MW314038
	PS-1479	*Citrus sinensis* (roots)	Alcanar, Tarragona, Spain	MW314313	MW314039

**Table 2 pathogens-11-00826-t002:** *Phytophthora* species isolated from plant tissues, floating leaves from two nursery ponds, and water samples taken from the irrigation system in the surveyed nurseries.

*Phytophthora* spp.	Host	Nursery	Source	N. Samples	Region
BIL		13		1	
CAC	*Fagus sylvatica, Photinia* “Red Robin”, *Pinus pinea*	6, 7, 9, 10, 12, 13, 19	R, S, W	14	Cat., Com. Val., Bas. C.
CAM	*Quercus ilex*	3, 7, 9	R, W	9	Com. Val.
CIN	*Arbutus unedo, Pinus radiata, Pseudotsuga menziesii*	9, 18	R, S	4	Cat., Bas. C.
CIP	*Citrus sinensis, Escallonia* sp., *Picea pungens* “Glauca Globosa”, *Quercus faginea, Rosmarinus officinalis*	4, 10, 11, 13, 15, 16	R, S, W	9	Cat., Com. Val.
CRA	*Pinus pinea*	17	S	3	Ext.
HED	*Juniperus phoenicea, Q. ilex, Viburnum tinus*	5, 7, 8, 10, 17	R, S	7	Cat., Com. Val., Ext.
GON	*Juniperus communis* “Hibernica”, *Q. ilex*	2, 8, 9, 10, 12, 13	R, S, L, W	9	Cat., Com. Val.
LAC		2, 8, 10	L, W	8	Cat.
NIC	*Buxus sempervirens, Citrus sinensis, Escallonia* sp., *Myrtus communis* “Tarentina”, *Pistacia lentiscus, Rosmarinus* sp.	7, 10, 15, 16	R	6	Cat.
NIE	*Arbutus unedo*	24	S	1	Ext.
PAL	*Cupressus sempervirens, Pistacia lentiscus*	7, 13, 16	R, W	3	Cat.
PLU	*Chamaecyparis lawsoniana* “Elwoodii”, *Cupressus sempervirens, Juniperus chinensis* “Expansa”, *Q. faginea, Q. ilex*	2, 4, 9, 10, 12, 16	R, S, W	14	Cat., Com. Val.
PSC	*Chamaecyparis lawsoniana* “Elwoodii”, *Q. ilex, Yucca rostrata*	2, 8, 10, 12,15, 23	R, S, W	17	Cat., Com. Val., Ext.
SAN	*Quercus ilex*	2	R, S	3	Com. Val.
TRO	*Arbutus unedo, Juniperus communis* “Hibernica”	13, 23	R	2	Cat., Ext.
SP. 1	*Citrus sinensis*	16	R	3	Cat.

BIL: *P. bilorbang*. CAC: *P. cactorum*. CAM: *P. cambivora*. CIN: *P. cinnamomi*. CIP: *P. citrophthora*. CRA: *P. crassamura*. HED: *P. hedraiandra*. GON: *P. gonapodyides*. LAC: *P. lacustris*. NIC: *P. nicotianae*. NIE: *P. niederhauserii*. PAL: *P. palmivora*. PLU: *P. plurivora*. PSC: *P. pseudocryptogea*. SAN: *P. sansomeana*. TRO: *Phytophthora* sp. tropicalis-like 2. SP. 1: *Phytophthora* sp. 1 clade 2. R: roots. S: soil. L: leaves. W: water from irrigation ponds in the nursery. Cat.: Catalonia. Com. Val.: Comunidad Valenciana. Ext: Extremadura. Bas. C.: Basque Country. Each nursery was coded with a number.

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
