# Peer review of "New Reports of Phytophthora Species in Plant Nurseries in Spain"

_pathogens, 2022, doi:10.3390/pathogens11080826_

Round 1
Reviewer 1 Report
Please see attached file

Reviewer 2 Report
Please see my recommendations for the manuscript below.
Please add the names of the sampling sites in Figure 1. This may be done by adding a legend or simply typing the site name alongside the bullets.
May I please ask why apple baiting was exclusively used for both soil and water samples? I guess a good strategy would have to use both apple and leaf baiting? Do you think this could have impacted the recovery of Phytophthora species?
Please use more than one approach for your phylogenetic analyses, since this is now standard practice for phylogenetic analyses. I recommend that the authors combine ML and Bayesian approaches with their MP, or at least another one. Aside from that, the resolution of the tree is quite low. Hence, I propose that the authors incorporate the high-resolution image. Please also highlight the isolates from this study in bold type.
All the alignments should be submitted to TreeBase and the accession number should be included.
At least one isolate of each of the Phytophthora species should be deposited in a culture repository. If this is not feasible, then at least the isolates of the putative new species should be submitted and the collection number should be included in the manuscript.
Why is the mention of fungal isolation and listing fungal species important for this manuscript? The focus of the manuscript should be Phytophthora or Oomycete. Hence, I would kindly suggest the authors remove the fungal part and adjust the manuscript accordingly.
The discussion, in my opinion, should be extensively revised because it comprises a substantial quantity of literature study. This is a research article rather than a review. Hence, the discussion should centre on the study's objectives. There is also a good scope to reduce redundancy. For example, a paragraph has been allocated to each new species identified in Spain. I believe they can all be merged into a single paragraph.
L350-358 None of the specific epithets are italicised in this paragraph.
L359 One sentence can't be a paragraph.
Reviewer 3 Report
In this study, the authors conducted a survey in plant nurseries for Phytophthora species. It is an interesting and useful study. The study involves microbe isolation and identification through sequencing and phylogenetic analysis. A plant health related journal or a taxonomic journal could be a better fit for the study.
Please see my comments below:
Introduction
Line 34-45, the statement can be stronger if major diseases caused by the species were mentioned.
Line 41 and 44, Phytophthora ramorum was mentioned twice.
Line 49, reference is needed for the inocula increase. A specific example should be provided here.
Line 50, Polycyclic diseases…
Line 46 to 51. The paragraph needs re-written. It started with pathogen movement, but then talked about inoculum multiplication.
Line 52-59. The description in this paragraph lacks examples. There are plenty of research studies on Phytophthora spp., but no specific examples were mentioned here. And the concept description is vague.
Line 60-72. What is the evidence of the increase in Phytophthora species diversity? Maybe they were just not reported locally.
Line78-84. Seems that fungal pathogens were also studied, but there was not much introduction of fungal pathogens in the Introduction.
Materials and Method
Line 144. What would be considered as potential Phytophthora and Pythium? What was the basis?
“Apple slices as baits” How did you sanitize apple slices and make sure that the apple slices did not carry any microbes?
Results
Discussion
The second half of the discussion part was not relevant to this study. With a survey, the authors cannot conclude the how the pathogens were dispersed and imply management. If a pathogen is already established in an area, cultural control may not be able to successfully management the pathogen.
Line 350 to 358. Latin names were not italicized.
Line 350-378. These paragraphs listed Phytophthora species reported in other regions. What does this imply?
Line 403. What does “Cylindrocarpon-like asexual morphs” mean?
Line 405-406. Reference is needed here.
Line 406-408. What does this sentence imply? Why Cylindrocarpon-like asexual morphs were studied and analyzed?
Line 409-414. How does this paragraph be implied from your study?
Line 415-420. Can you conclude the pathogen movement methods from your study?
Fungal pathogens were also isolated from the study, but their role were not discussed much.
Multiple microbes were isolated from the same plant, how could you prove that it was a Phytophthora sp. causing the infection?
The title was “new reports of ..”, but not all Phytophthora spp. isolated in this study were new. In order for you to consider the Phytophthora species as new report of pathogens, pathogenicity assay should be conducted.
Round 2
Reviewer 2 Report
Dear Authors, Thank you for kindly addressing my suggestion for the previous version of this manuscript. I have a few suggestions on the figures that I have listed below.
Figure 1. Thanks for adding the sites names on the map. However, it is challenging to read these names because they are in purple/pink font. I will suggest the authors use a black font. I guess there is no need to add the province name along with the sampling site because it makes the illustration busy. Provincial details can be added to the figure legend. Catalonia, for example, may be a light shade of blue, and then you specify blue=Catatonia in the figure legend.
Figure 2. Please remove the shadow from the bars on the plots. The best strategy is to make them solid colours and a bit wider than what they are now. In doing so, you can place the numbers inside the bar and not on the top. Please also italicise Phytophthora in the legend.
The phylogenetic tree in Figure 3 is of poor resolution. Indicating the species identified in the present study in bold font should be enough. So, please remove the underlines. All the taxon names should be italicised, such as P. cinnamomi, P. cactorum and many others.
Reviewer 3 Report
In this study, the authors conducted a survey in plant nurseries for Phytophthora species. It is an interesting and useful study. The study involves microbe isolation and identification through sequencing and phylogenetic analysis. A plant health related journal or a taxonomic journal could be a better fit for the study.
Please see my comments below:
Line 38 and 41, Phytophthora ramorum was mentioned twice.
Line 43 to 48. The paragraph needs re-written. It started with pathogen movement, but then talked about inoculum multiplication.
Line 139. How did you identify Phytophthora and Pythium cultures? What was the basis?
Line 124-136. “Apple slices as baits” How did you sanitize apple slices and make sure that the apple slices did not carry any microbes?
Line 384. You did not mention any results about “Cylindrocarpon-like”, so how many Cylindrocarpon-like did you isolate?
